# Does entropy modulate the prediction of German long-distance verb particles?

**Kate Stone**[1]*, **Shravan Vasishth**[1], **Titus von der Malsburg**[2]

**1** Department of Linguistics, University of Potsdam, Potsdam, Brandenburg, Germany, **2** Institute of Linguistics, University of Stuttgart, Stuttgart, Baden-Württemberg, Germany

* stone@uni-potsdam.de

**Data Availability Statement:** All data, code, and materials are openly available at: https://osf.io/h75jm/ DOI 10.17605/OSF.IO/H75JM.

**Funding:** Kate Stone was supported by a doctoral scholarship from the Potsdam Graduate School.

## Abstract

In this paper we examine the effect of uncertainty on readers' predictions about meaning. In particular, we were interested in how uncertainty might influence the likelihood of committing to a specific sentence meaning. We conducted two event-related potential (ERP) experiments using particle verbs such as *turn down* and manipulated uncertainty by constraining the context such that readers could be either highly certain about the identity of a distant verb particle, such as *turn the bed [. . .] down*, or less certain due to competing particles, such as *turn the music [. . .] up/down*. The study was conducted in German, where verb particles appear clause-finally and may be separated from the verb by a large amount of material. We hypothesised that this separation would encourage readers to predict the particle, and that high certainty would make prediction of a specific particle more likely than lower certainty. If a specific particle was predicted, this would reflect a strong commitment to sentence meaning that should incur a higher processing cost if the prediction is wrong. If a specific particle was less likely to be predicted, commitment should be weaker and the processing cost of a wrong prediction lower. If true, this could suggest that uncertainty discourages predictions via an unacceptable cost-benefit ratio. However, given the clear predictions made by the literature, it was surprisingly unclear whether the uncertainty manipulation affected the two ERP components studied, the N400 and the PNP. Bayes factor analyses showed that evidence for our a priori hypothesised effect sizes was inconclusive, although there was decisive evidence against a priori hypothesised effect sizes larger than $1\mu V$ for the N400 and larger than $3\mu V$ for the PNP. We attribute the inconclusive finding to the properties of verb-particle dependencies that differ from the verb-noun dependencies in which the N400 and PNP are often studied.

## Introduction

Contextual cues in a sentence preactivate words in memory, such that processing of those words begins before they are seen [1, 2]. If contextual cues are sufficiently constraining, preactivation may crystallise into prediction of a specific word. Such lexical predictions can benefit readers in processing long-distance dependencies, where key information to interpreting an

The funders had no role in study design, data collection and analysis, decision to publish, or preparation of the manuscript.

**Competing interests:** The authors have declared that no competing interests exist.

event is delayed [3–5]. Take for example the German particle verb construction in (1), where a large amount of material separates the verb "fuhr" *carried* from its particle "fort" *on*:

(1). Der Professor **fuhr** mit seinem Vortrag trotz regelmäßiger
The professor **carried** with his lecture despite regular
Störungen **fort**.
interruptions **on**.

Rather than waiting for the particle to appear, and thus waiting to know the exact meaning of the sentence, readers may predict "fort" in order to facilitate processing of the intervening material. However, other plausible particles could include "ab" (*the professor drove off*), "zusammen" (*the professor was startled*), or "zurück" (*the professor reversed [his position] or drove back*). This creates uncertainty about the intended meaning of the sentence. In this paper, we use German particle verbs to test how uncertainty may affect predictive processing difficulty in long-distance dependencies. Although we did not find a conclusive answer to this question, we report our attempt as a resource for future researchers who may wish to revisit the issue. Furthermore, by reporting the inconclusive results, we avoid contributing to publication bias via the file drawer problem [6].

## Prediction in long-distance dependencies

Evidence that readers make long-distance predictions is most often observed for syntactic information, including in verb-particle constructions. Piai et al. (2013) found evidence that Dutch verbs that could take a particle elicited a larger left anterior negativity (LAN) than those that never took a particle. The LAN has been associated with maintaining information in working memory [7], suggesting that the verb triggered a prediction for a particle. However, there was no evidence that the number of different particles licensed by the verb affected LAN amplitude, suggesting that the prediction was for the syntactic features of the verb-particle dependency and not for its lexical properties. The LAN effect was still present at the direct object, suggesting that readers maintained their syntactic prediction while processing the intervening sentence.

Further evidence for long-distance syntactic predictions comes from filler-gap dependencies such as *I know what$_i$ you hit and broke the mirror with __$_i$* [8]. Encountering a transitive verb within a *wh*-dependency is argued to induce active prediction of a gap site, inducing longer reading times in the region of *hit* than at the same point in the non-*wh*-dependency sentence *I know that you hit and broke the mirror with the ball* [8–11]. This predicted gap is assumed to be maintained until it is resolved. Evidence from event-related potentials (ERPs) supports predicted gap maintenance by showing that a sustained anterior negativity (SAN) is larger for *wh*-questions that are expected to be long vs. short; this has been attributed to a higher burden placed on working memory when a longer structure is expected [12–14].

Maintenance of a syntactic prediction should facilitate downstream processing, and indeed such facilitation has been observed in reading times. For example, encountering the word *either* facilitates processing of a co-ordinate *or* structure in sentences like "The team took either the train or the subway. . ." compared to "The team took the train or the subway. . ." [15, 16]. This suggests that *either* triggers a syntactic prediction for the co-ordinate structure, which is sustained to facilitate reading at its resolution. Staub and colleagues also observed that syntactic predictions facilitated processing by protecting against misanalysis of S-coordination as NP-coordination. When *either* was present in sentences such as "Either Linda bought the red car or her husband leased the green one", the NP *her husband* was never misanalysed as the direct object of *Linda bought*. However, when *either* was absent, a strong preference for NP-

coordination appeared to trigger misanalysis, as suggested by a larger number of regressions in the spillover region following *her husband*. Long-distance syntactic predictions may therefore facilitate processing through both preactivation and protection against misanalysis.

In contrast to syntactic prediction, lexical prediction may represent commitment to a more specific sentence interpretation. Specific predictions can increase the facilitatory advantage of a correct prediction, but also run an increased risk of costly misprediction. The increased risk may explain why Piai and colleagues (2013) did not find ERP evidence of lexical prediction in their particle verb stimuli, as discussed above.

Commitment to a specific sentence interpretation refers to one account of the predictive mechanism in sentence processing where a specific interpretation might be stored in working memory [1, 2, 17, 18]. However, the mechanisms underlying predictive processing are by no means resolved. An alternative account is that the level of preactivation alone is sufficient to explain predictability effects [19]. This is supported by the success of metrics such as surprisal that measure the probability of a word given its context—which could be assumed to be linked to word preactivation—in accounting for variability in predictability-related empirical effects [20–25]. On the other hand, it has been shown that suprisal alone is not sufficient to explain all differences in reading times or the N400 [26, 27]. In the case of reading times, this was taken to suggest the need of an additional explanatory factors such as reanalysis or revision [26]. Assuming that reanalysis or revision can only be triggered if a commitment has been made, such findings could be taken as evidence that predictions can involve commitments.

On the other hand, reading time studies do provide some support for long-distance lexical predictions. A study of particle verb-like constructions in Persian found that increased lexical predictability of a head-final verb facilitated reading times in both short- and long-distance dependencies [4]. One interpretation of these results is that readers predicted the verb early in the sentence and sustained this prediction until it was resolved at the verb site. However, the pre-verbal information in both short- and long-distance conditions may have been informative about the identity of the verb, and so it is possible that the verb in this study was only predicted immediately prior to its being seen.

In similar constructions in Hindi, increased lexical predictability was also found to facilitate reading times at both short and long distance [3]. The pre-verbal region in this study was not informative about the identity of the verb, and so it could reasonably be concluded that the predictability effect in the long-distance condition was evidence of long-distance lexical predictions.

In support of the Hindi findings, a study of German particle verbs found that higher certainty about the identity of a particle facilitated eye fixation times at the particle in both short- and long-distance conditions [5]. In this study again, the pre-particle region was not informative about the identity of the particle, supporting the conclusion that long-distance predictions may have been made. However, the uncertainty effect was not replicated in self-paced reading times, suggesting that the effect of lexical certainty on eye movements may have been quite subtle.

Taken together, the above studies suggest that long-distance predictions are generated and maintained to facilitate the processing of long-distance dependencies. However, whether lexical content is included in these predictions appears to be more difficult to observe. One possibility is that higher variability among lexical versus syntactic continuations of a sentence creates a higher degree of uncertainty for the parser. Combined with the likelihood that intervening material may require the long-distance lexical prediction to be reanalysed, lexical predictions may be discouraged in all but the highest certainty situations. Even the potential facilitatory benefit of a correct prediction to processing the predicted word and the intervening material may be outweighed by potential reanalysis cost.

## The cost of uncertainty in predictive processing

The probability of a misprediction increases as uncertainty about a sentence increases, making uncertainty a potentially important factor in whether readers make predictions about long-distance dependencies. A reader's uncertainty at a given point in a sentence can be measured with entropy, which quantifies the distribution of different ways a sentence could continue [20]; for example, via the number of different continuations provided at a particular point in the sentence in a sentence-completion task. A link between entropy and predictive processing cost has been observed in reading times. Encountering a word that triggers entropy among plausible sentence continuations has been associated with longer reading times, suggesting that uncertainty creates processing cost via the prediction and maintenance of multiple potential continuations in memory [28, 29]. Large reductions in entropy have also been associated with increased reading times, suggesting that large shifts in expectation also incur a processing cost [20, 24, 30, 31].

The relationship between entropy and processing cost is less well-studied in event-related potentials (ERP). Several experiments have examined the effect of contextual constraint on predictive processing, and constraint can be interpreted as a measure of entropy at a target region of a sentence. However, constraint in such studies is usually quantified using the highest probability word given at the target region in a sentence-completion task (a cloze test); for example, see [32–34]. The degree of entropy among different cloze test completions is usually not taken into account.

Entropy among cloze test completions is important if we assume that it represents memory activations in a single reader [35]. Consider a sentence where the most strongly activated word at a given point has a cloze probability of 36% and that two other activated words each have a probability of 32% (high entropy). A reader would be highly uncertain about how the sentence will continue, given there are three almost equally probable options. Compare this to a sentence where the strongest activated word also has a probability of 36%, but where there is a large spread of other activated words each with a probability of no more than 5% (lower entropy). The reader can now be relatively certain about how the sentence will continue, even though the target word has the same probability as in the first sentence. Under the highest-cloze-probability definition of constraint used in many ERP studies, both sentences would be grouped together as weakly constraining, even though the difference in uncertainty will have different consequences for predictive processing.

The highest-cloze-probability definition of constraint may explain inconsistency among ERP findings about the effect of constraint on processing cost. In particular, studies have observed a larger anterior late positivity (PNP) for unexpected words in strongly versus weakly constraining contexts, suggesting that higher certainty leads to predictions which are more costly when disconfirmed [32, 33, 36]. The PNP has been linked to the suppression of disconfirmed sentence representations, potentially shedding light on the mechanism driving the processing costs observed in reading times [37]. However, not all studies of the PNP find an effect of constraint, or have observed it in the opposite direction [34, 38–42].

Few studies have directly tested entropy in ERP, as opposed to constraint. In an exploratory analysis of Thornhill and Van Petten's (2012) data, a reduction in entropy at the target word was found to explain some variance in the PNP, although this did not survive a confirmatory analysis [23]. In Piai et al.'s (2013) study of Dutch particle verbs, verbs that took large number of particles were not found to elicit differences in the left anterior negativity (LAN) compared to verbs that took a small number of particles. While entropy among particles was not explicitly quantified, the findings can be interpreted as showing that differences in lexical uncertainty did not result in differences in working memory burden, as indexed by the LAN. The

processing cost of disconfirmed predictions in relation to entropy was not investigated. Given that uncertainty may be a deciding factor in predictive processing of long-distance dependencies and that ERP has the potential to reveal mechanisms driving processing cost, we conducted two ERP experiments examining the effect of entropy on predictive processing.

## The current experiments

The aim of the experiments was to test whether lexical entropy would affect readers' willingness to predict a long-distance verb particle. We compared sentences which required a particle to be semantically plausible, but varied in how many different particles could complete the sentence. Since we were specifically interested in the effect of entropy among the different particles' lexical representations, it was strictly controlled: In the example experimental item (2) below, only a single particle was semantically plausible in conditions (a) and (b). In conditions (c) and (d), at least two particles were plausible. We hypothesised that readers should experience more uncertainty about the meaning of the sentence in (c/d), and thus be less likely to predict a specific particle given the increased potential for a misprediction and its associated cost:

(2). **1 plausible particle, expected**:

 a. Der ordentliche Professor **fuhr** mit seinem Vortrag trotz
 The orderly professor **carried** with his lecture despite
 regelmäßiger Störungen immer ordnungsgemäß **fort**, da er für
 regular interruptions always properly **on**, as he for
 seine Unaufgeregtheit bekannt war.
 his unflappability known was.
 **1 plausible particle, violation**:

 b. *Der ordentliche Professor **fuhr** mit seinem Vortrag trotz
 The orderly professor **carried** with his lecture despite
 regelmäßiger Störungen immer ordnungsgemäß **mit**, da er für
 regular interruptions always properly **with**, as he for
 seine Unaufgeregtheit bekannt war.
 his unflappability known was.
 **2+ plausible particles, expected**:

 c. Der ordentliche Buchhalter **fuhr** seinen zuverlässigen
 The orderly accountant **turned** his reliable
 Computer bei der Arbeit immer ordnungsgemäß **herunter/hoch**,
 computer at work always properly **off/on**,
 da er für seine korrekte Arbeitsweise bekannt war.
 as he for his correct work practices known was.
 **2+ plausible particles, violation**:

 d. *Der ordentliche Buchhalter **fuhr** seinen zuverlässigen
 The orderly accountant **turned** his reliable
 Computer bei der Arbeit immer ordnungsgemäß **mit**, da er für
 computer at work always properly **with**, as he for
 seine korrekte Arbeitsweise bekannt war.
 his correct work practices known was.

We used the N400 and the post-anterior positivity (PNP) to measure processing cost at the particle. In particular, we were interested in ERP evidence of processing cost when an

unexpected, semantically implausible particle was encountered in conditions (b) and (d). We focused our inference on these conditions because the target particles both had a cloze probability of zero. This allowed us to rule out the possibility that any ERP difference observed at the particle was due to differences in cloze probability, which correlate strongly with amplitude of the N400 and the PNP [33, 43]. While cloze probability is a measure of a word's predictability, it does not necessarily equate to whether a word is predicted: Preactivation of a word by its context—which increases cloze probability [35]—also makes that word easier to access in the lexicon and integrate into the building sentence representation once it is seen, without it having been specifically predicted. We assume a difference between preactivation and prediction, where preactivation can reflect activation of several different items in memory, while prediction reflects commitment to one specific item, for example by adding it to working memory [1, 2, 17, 18]. Differences in ERP amplitude associated with a target word's cloze probability may therefore also be driven by non-predictive processes triggered by having seen that word. In contrast, a difference between two target words with the same cloze probability but differing in entropy—the distribution of words not seen but preactivated—should be informative about predictive processing. The target words in (b) and (d) were always identical, and the pre- and post-critical regions were matched between conditions to rule out any influence of the preceding or following words on the ERP. The pre-critical region was not informative about the identity of the particle.

We use the term "zero probability" when referring to the implausible particles to reflect the fact that the sentence context is highly unlikely to preactivate implausible particles in memory as they would be meaningless in the context. However, there exists evidence from the N400 demonstrating that readers can be sensitive to differences within the very low end of the probability spectrum, including for implausible words [25]. These activations are unlikely to be captured by a cloze test and thus have a cloze probability of zero, but their preactivation may still affect ERP amplitude. This would be a caveat to our "matched zero probability" target particles. The contribution of very low probability words to entropy and ERP amplitude needs future investigation, but for the current study we use the term "zero probability" to refer broadly to a situation where, for the purposes of studying the strength of people's probabilistic representations of the context of a sentence, the contribution of preactivation from an implausible particle is effectively zero.

To detect the cost of violated predictions, we examined two ERP components: the N400 and the anterior post-N400 positivity (PNP). The N400 is a negative deflection in the ERP occurring at around 250–500 ms after a word is seen and is highly correlated with a word's probability given the preceding sentence context [43]. Amplitude of the N400 decreases at each new, context-congruent word in a sentence [44, 45] and increases at any unexpected word, inversely proportional to its probability [43, 46]. This has led to the hypothesis that sentence context allows the preactivation of probable words that are then easier to process once encountered, and that the N400 reflects the lexical access or update cost incurred by words with low preactivation [43, 47].

Previous studies involving uncertainty have tested the effect of contextual constraint on the N400, but have not found that constraint affects its amplitude when the cloze probability of the target words is matched [32, 33, 41, 46, 48]; but see [49, 50]. That is, two words with the same low probability will trigger N400s of the same amplitude, even if one of them is found in a strongly constraining context and the other in a weakly constraining context. Lexical access and probabilistic update models of the N400 explain this as occurring because each low probability word has received no preactivation, meaning that lexical access is equally unfacilitated [43] or that the shift in model update is equally large [47]. This presents a testable null hypothesis for the current study: if the N400 is not affected by constraint—which should reflect

entropy—then we should not expect to see a difference in N400 amplitude between 1- and 2+particle violations. However, given that few previous studies of the N400 have directly tested entropy, it is possible that we could see a larger N400 in the 1-particle condition reflecting greater update or more difficult lexical access after a misprediction.

The PNP is a positive amplitude deflection in the anterior region of the scalp in the 600–900 ms window [32, 33, 46]. The PNP appears to be functionally distinct from the more well-known posterior P600 [32, 33], which has been associated with conflict detection and repair processes [51–56]. In contrast to the P600, the PNP has been associated with suppression of disconfirmed sentence representations [33, 37] and appears only to be sensitive to plausible but unexpected words, suggesting that suppression may only occur if there is a viable, alternative way to update the event representation [32, 33, 57]. If this is the case, we may not expect to see a PNP modulation by our implausible particles.

However, in addition to inconsistent findings about the effect of constraint on the PNP already mentioned [34, 38–42], the plausibility feature has only ever been tested in verb-noun dependencies. A context-violating noun may be implausible at multiple levels of representation (e.g. animacy, thematic role, semantic), whereas our implausible particles present a purely semantic violation. It is thus less clear how the PNP might be affected. If purely semantic violations can elicit the PNP, we may see a higher cost (larger PNP) of having to suppress a disconfirmed sentence representation at the violation particle in the 1-particle condition than in the 2+particle condition. If not, then we may see no difference associated with entropy if the PNP is not triggered by the implausible violations.

We tested these hypotheses in two ERP experiments: in Experiment 1, we used the $2 \times 2$ design set out in example (2) to compare the effect of the implausible particles on the N400 and PNP, but also to confirm that the particles could elicit an N400 and PNP in the implausible versus plausible conditions. In Experiment 2, we attempted to replicate the findings of Experiment 1 using a simpler design with a larger number of participants and experimental items. Data, code, and materials for both experiments can be found at https://osf.io/h75jm/.

## Experiment 1

Experiment 1 tested whether entropy would affect readers' predictions about an upcoming verb particle. We inferred that predictions had been made by measuring processing cost at prediction violations. We predicted that a larger N400 and PNP would reflect the greater cost of recovering from a mis-prediction in the 1-particle condition than in the 2+particle condition where a prediction was less likely. Experiment 1 was pre-registered at https://osf.io/qbna2; see also S1 Appendix.

### Materials and methods

**Participants.**   Fifty-four participants were recruited. The sample size was determined by recruiting as many participants as was possible during one university semester. Four participants were excluded due to a large number of their target EEG trials being contaminated by muscle and/or blink artefacts. This left a total of 50 subjects (6 male), with a mean age of 25 years (range: 17–40 years, *SD* = 5 years). In line with university policy, all participants were reimbursed for their time either financially or in the form of credit points toward their studies. All participants were right-handed German native speakers, with no known history of developmental or current language, neurological, or psychiatric disorder, and had not participated in the cloze test. The study was conducted in line with the principles of the Declaration of Helsinki and all participants provided written consent to participation in the study. In accordance

with German law, IRB review was not required as the study involved only healthy adult participants.

**Materials.** For each particle verb, two sentences were constructed, as seen in Example (2). The position of the base verb and particle in each sentence pair was matched. The particle was the target word. Within each sentence pair, at least two words before the particle were identical. 103 sentence pairs were constructed and presented as a cloze test to 30 German native speakers (mean age 25 years, SD 6 years, range 18–41 years) on a desktop computer in our in-house lab using the Ibex software [58]. The sentence pairs were divided into two lists, such that each participant only saw one condition from each item. The particle of the sentence was replaced by a gap, which participants were asked to fill with the first word that came to mind.

The items were then ranked in terms of how well they fulfilled the criteria that the 1-particle condition elicited only one particle and the 2+particle condition elicited at least two particles with similar probability. To rank the items, cloze probabilities were calculated for each particle completion. Other kinds of completions were grouped into categories (e.g. prepositional phrases, adjectives, nouns) and a cloze probability was calculated for each category. Items were then ranked by entropy among the responses (lowest to highest in the 1-particle condition; highest to lowest in the 2+particle condition). For the 2+particle condition, further weight was given to items where the two highest ranked particles were close in cloze probability. This ranking scheme left a final set of 40 plausible items fulfilling the criteria of the experiment. Cloze probability and entropy statistics are summarised in Table 1.

To create the violation conditions, two German native speakers selected particles that were not integrateable into the sentence context, including illicit verb-particle combinations. No participant in the cloze test produced any of these illicit particles. The same particle was used in both violation sentences within each item. The 40 critical sentences used in the analysis were split into four lists in a Latin square design, such that each participant only saw one of the four conditions for each critical item and thus a total of 10 items per condition. Critical sentences were pseudo-randomly interspersed with 98 filler sentences. Sixty-two filler sentences contained plausible sentences in a variety of lengths and tenses, some of which contained non-separated particle verbs (e.g. hochgefahren, *switched on*). The remaining 36 fillers were particle verb sentences from the cloze test that did not fulfill the 1- vs. 2+particle criteria. In total, each participant saw a total of 138 sentences. The ratio of plausible to implausible sentences was approximately 3:1. The order of presentation of sentences within each list was pseudo-randomised via the presentation software. Each sentence was followed by a yes/no question appeared that probed different regions of the sentence; for example, the question for example (2) above was "Verlief der Vortrag ungestört?" *did the lecture proceed uninterrupted?*.

**Procedure.** Participants sat in a shielded EEG cabin approximately 60 cm from a 56 cm presentation screen. The experimental paradigm was built and presented using Open Sesame [59]. Each experimental session began with an instruction screen advising participants that they would read sentences presented word-by-word and that after each sentence, they would

**Table 1. Cloze probability and entropy summary statistics for Experiment 1.**

| Condition | Cloze probability target particle Mean [95% CrI] | Difference between 1st- and 2nd-best completion Mean [95% CrI] | Entropy target particle Mean [95% CrI] |
|---|---|---|---|
| 1-particle | 0.90 [0.74, 1.00] | 0.73 [0.64, 0.85] | 0.32 [0.31, 0.33] |
| 2+particle | 0.54 [0.53, 0.56] | 0.29 [0.16, 0.39] | 0.81 [0.80, 0.81] |

1st- and 2nd-best completions refer to the highest and second-highest cloze particles at the target site.

answer a question using a video game controller. Participants were instructed to answer as quickly and accurately as possible. Each experimental session began with four practice trials.

Each trial in the experiment began with a 500 ms fixation cross in the centre of the screen followed by a blank screen jittered with a mean of 1000 ms and standard deviation 250 ms. Each sentence was presented word-by-word for a duration of 190 ms per word plus 20 ms for each letter. The target word was always presented for 700 ms regardless of length. The inter-word interval was 300 ms. A comprehension question appeared after each sentence and was answered via the video game controller, which triggered the next trial. Breaks were offered after every 30 sentences. The testing session including EEG setup lasted approximately two hours.

**EEG recording and preprocessing.** The EEG recordings were made in the Department of Linguistics at the University of Potsdam, Germany, in a purpose-built, electro-magnetically shielded EEG cabin using a 32-lead system and electrodes arranged on the head based on the international 10–20 system [60]. Electrode impedances were kept below $10k\Omega$ throughout the experiment. EEG was recorded at a sampling rate of 512 Hz and online filtered with a low-pass filter of 138 Hz and using the left mastoid as a reference.

Raw EEG recordings were downsampled offline in BrainVision Analyzer 2, Version 2.1.2, to 500 Hz for ease of processing and interpretation. Zero phase shift IIR Butterworth filters were applied with a low-pass cut-off at 0.01 Hz (order of 2, time constant of 15.92) and a high-pass cutoff at 30 Hz (order of 2, no time constant). A notch filter was applied at 50 Hz. The full recording was then segmented into epochs from sentence onset to question onset. Ocular correction using restricted Infomax was applied to the sentence epochs using automatic independent component analysis (ICA) with a meaned slope algorithm. The reference electrodes were two electrodes placed at the left outer canthus and above the left eye to record horizontal and vertical eye movements, respectively. The bound number of blinks was 60 with a convergence bound of $10^{-7}$. The number of ICA steps was 512. Components were found using sum of squared correlations with the horizontal and vertical ocular electrodes [61–63]. The total value to delete was 30%.

The EEG was then re-referenced to the average of the two mastoids. The corrected segments were further segmented into 1200 ms epochs representing a period of 200 ms before the onset of the target word (the particle), and 1000 ms after onset. EEG segments with muscle artefact or irreparable eye-blink or movement artefact were automatically marked for 200 ms before and after each respective artefact, defined as exceeding:

1. a maximum voltage step of more than 50 $\mu V$

2. a maximum absolute difference of 200 $\mu V$ in a 100 ms interval

3. a minimum amplitude of -100 $\mu V$

4. a maximum amplitude of 100 $\mu V$

5. a minimum low activity of 1 $\mu V$ in a 100 ms interval.

Marked segments were then visually inspected and discarded if they indicated muscle artefact or a technical issue. This resulted in the exclusion of 6.20% of the 1000 target trials used in the statistical analysis. A further 0.50% of trials were excluded due to question response times over 10 seconds (indicating a technical problem) and 0.30% were not recorded due to experimenter error. The data were then exported and baseline-corrected in R using the package *eeguana* [64].

**Analysis.** A linear mixed effects model with full variance-covariance matrices to account for the individual variability of subjects and items was fit to data from the violation conditions

(b) and (d) using the *brms* package for R [65]. The dependent variable for the N400 was mean amplitude across the electrodes Cz, Pz, CP1, and CP2 in the 250–500 ms time window. The dependent variable for the PNP was mean amplitude across the electrodes Fz, FC1, and FC2 in the 600–900 ms window. For Experiment 1, these regions were chosen post hoc after we deviated from the pre-registered single-electrode analysis (see S1 and S2 Appendices). The regions were generally consistent with the distribution of the N400 and PNP observed in the predictive processing literature [32, 33, 66], although some studies observe a more anterior PNP [33, 36]. Since they were chosen post hoc, the regions of interest in Experiment 1 were used as pre-registered regions in Experiment 2.

The predictor 'number of plausible particles' was effect contrast coded to reflect our expectation for larger ERPs at 1-particle violations: 0.5 (1-particle), -0.5 (2+particle). The decision to use a categorical predictor rather than raw entropy was made because the precise functional relationship between entropy and ERP amplitude was not known, and we felt unable to safely assume it was linear.

To constrain the model to plausible values and avoid overfitting, we placed regularising priors of $N(0, 1)$ on each parameter [67–70]. This prior with a normal distribution centered on zero indicates to the model that, for example, the effect of the number of plausible particles will most likely be small or zero, with either a positive or negative sign, and has a 95% probability of lying between $-2$ and $2\mu V$.

To quantify how much evidence we had for any N400 and PNP effect, we conducted Bayes factor analyses comparing models with the predictor number of particles (M1) versus a reduced model without this predictor (M0), i.e. $BF_{10}$. A Bayes factor of approximately 1 would indicate no evidence in favour of either model. As the Bayes factor increases over 3, evidence strengthens in favour of M1 [71, 72]. As the Bayes factor decreases under 0.3, evidence strengthens in favour of M0. Since the Bayes factor is sensitive to the choice of prior [73, 74], we conducted a sensitivity analysis by computing Bayes factors for a range of priors to determine how these affected our conclusions [75, 76]. We used a range of priors that assumed a priori effect sizes ranging from small, between $-0.2$ and $0.2\mu V$, to large, between $-10$ and $10\mu V$.

To confirm that the violation particles elicited an N400 and PNP relative to their plausible counterparts in line with previous studies [77], we fit the above models to all conditions (a-d) and added the predictor 'plausibility'. Plausibility was effect contrast coded to reflect our expectation for larger ERPs in the implausible conditions: 0.5 (implausible), -0.5 (plausible) and a Bayes factor was computed at a prior of $N(0, 1)$. Since the plausible particles differed in cloze probability and identity from each other and from the violation particles, we used this analysis as a sanity check only and did not conduct any further analyses.

## Results

Mean amplitude relative to the onset of the target particle is plotted in Fig 1.

In Fig 1A and 1B, the 250–500 ms window of the ERP shows a strong negative deflection over the posterior scalp for the implausible conditions relative to the plausible conditions, consistent with an N400. Implausible particles also elicited a large positive deflection in the 600+ ms time window relative to plausible particles over the posterior scalp, consistent with a P600.

Fig 1C and 1D shows a positive deflection in the post-N400 period of the ERP (i.e. after 500 ms) in fronto-central electrodes consistent with a PNP in the 1-particle condition relative to the 2+particle condition and relative to the plausible conditions.

**Analysis of the N400.**  At implausible particles, the N400 was more negative in the 1-particle condition than in the 2+particle condition and the mean of the posterior was $-0.34\mu V$ with

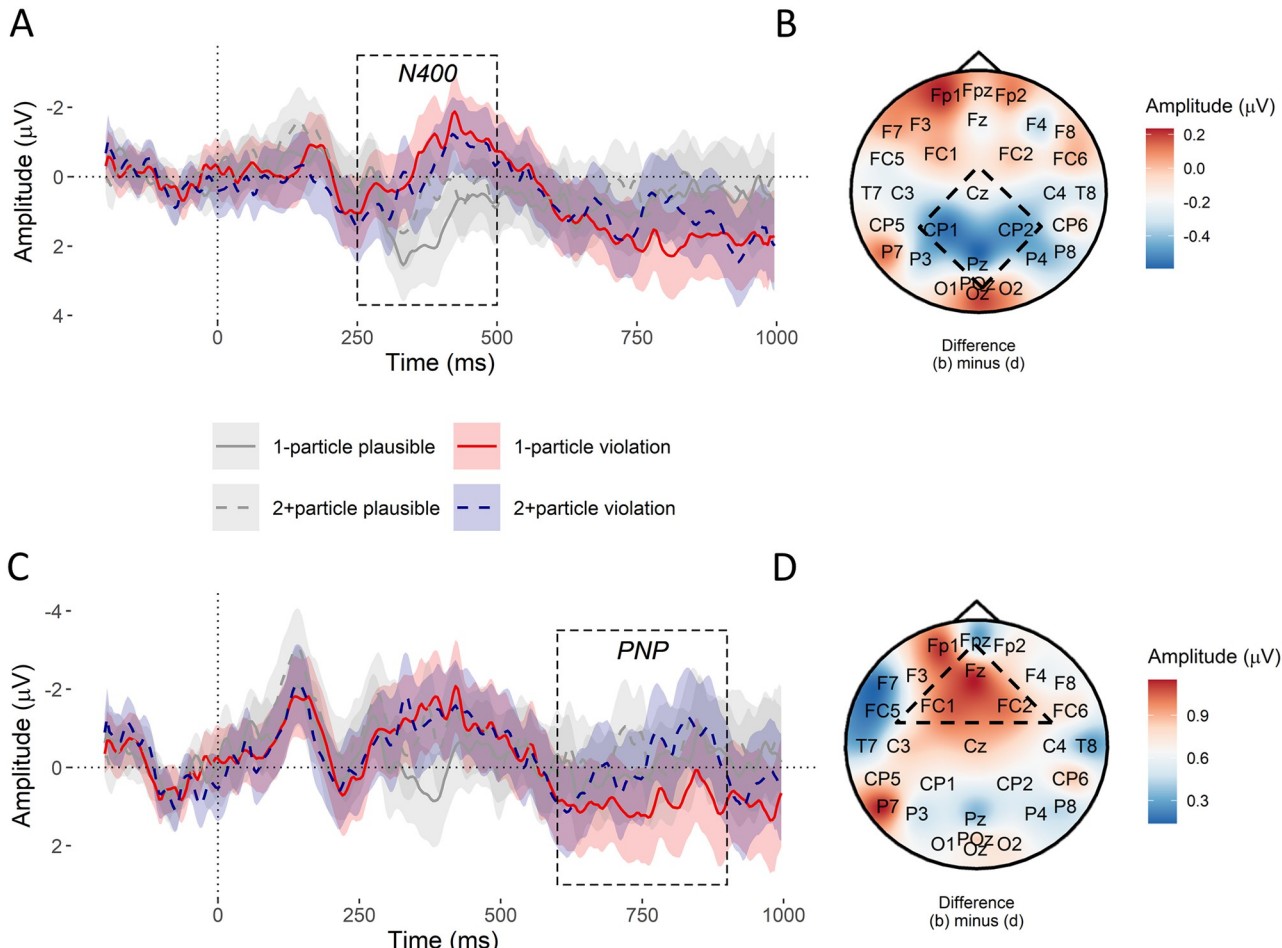

**Fig 1. Experiment 1: ERP results. A**. Average amplitude following stimulus onset shows an N400 in the posterior region of interest (Cz, CP1, CP2, Pz) for violations of both the 1-particle (red) and 2+particle (blue) conditions relative to the plausible (grey) conditions. Ribbons show 95% confidence intervals. These intervals were calculated by fitting a linear model of the form *amplitude ~ condition* with by-subject and by-item intercept adjustments at each timestep of the ERP recording and extracting the standard error of the relevant conditions. The analysed N400 time window is overlaid. **B**. A voltage subtraction map of the N400 time window shows a small difference in amplitude between the 1-particle (a) and 2+particle (b) conditions in the region of interest (dashed square). **C**. A PNP for the 1-particle condition is seen in the anterior region of interest (Fz, FC1, FC2) 600–900 ms window after stimulus onset, but not in the 2+particle or plausible conditions. Ribbons show 95% confidence intervals. **D**. A voltage subtraction map of the PNP time window suggests amplitudes in the 1-particle condition were more positive in the mid-frontal region (dashed triangle) than in the 2+particle condition.

a 95% credible interval (CrI) of $[-1.10, 0.39]$ $\mu V$ (Fig 2A). The Bayes factor at a prior of $N(0, 1)$ favoured neither the null or alternative hypotheses about an entropy-related difference in the N400, $BF_{10} = 0.57$. The sensitivity analysis suggested there was moderate evidence against a priori hypothesised large effect sizes, but was inconclusive about small effect sizes (Fig 2B). There was strong evidence that the N400 was more negative for implausible than plausible particles, $\hat{\beta} = -1.09 \, \mu V, [-1.69, -0.48], BF_{10} = 465$.

**Analysis of the anterior post-N400 positivity (PNP).** At implausible particles, the posterior mean indicated more positive amplitude in the 1- vs. the 2+particle condition, $\hat{\beta} = 0.74 \, \mu V$, 95% CrI $[-0.25, 1.77]$ $\mu V$ (Fig 2C). However, the Bayes factor at a prior of $N(0, 1)$ favoured neither the null or alternative hypotheses about an entropy-related effect on the PNP, $BF_{10} = 1.50$. The sensitivity analysis suggested effect sizes of approximately $1\mu V$ were

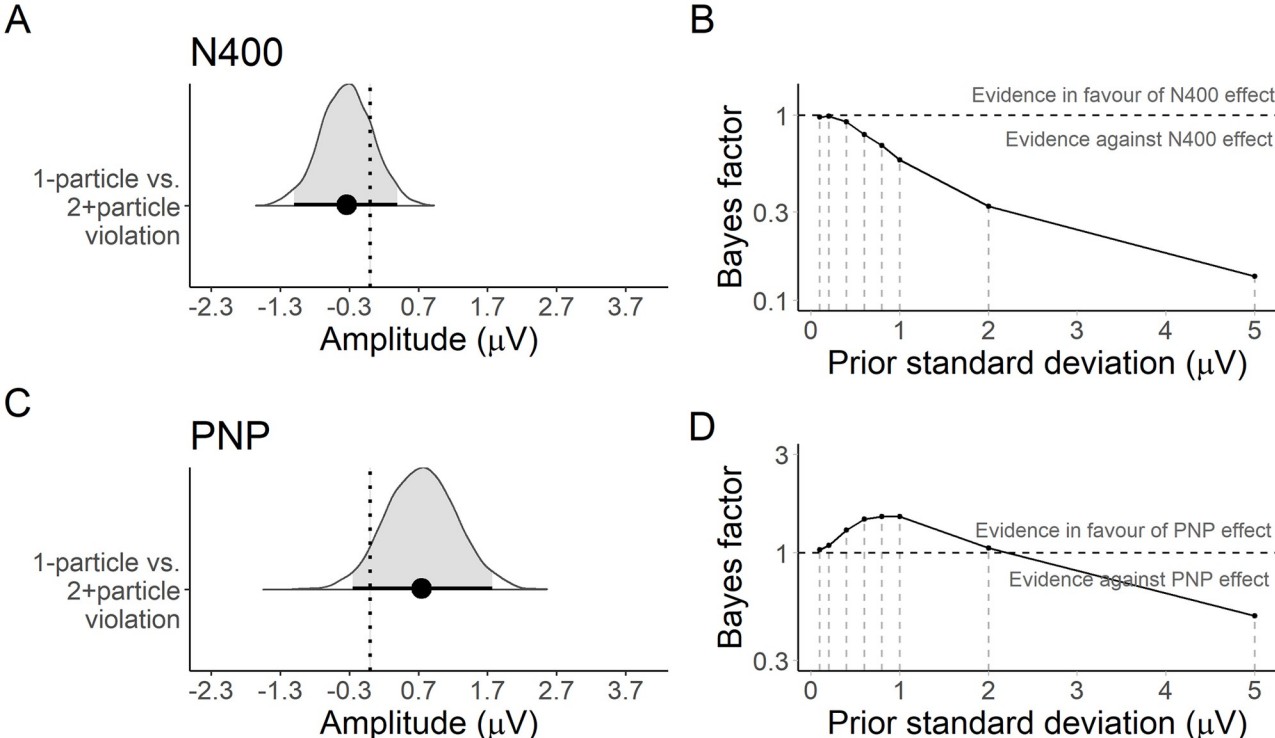

**Fig 2. Experiment 1: Posterior estimates and Bayes factor analysis for the planned and exploratory comparisons. A**. The posterior distribution represents the estimated change in N400 amplitude associated with violations in the 1-particle condition relative to the 2+particle condition. The point and errorbar reflect the posterior mean and 95% credible interval. **B**. Bayes factors computed under a range of priors comparing models with and without a predictor for the 1- vs. 2+particle condition manipulation. A Bayes factor of 1 indicates no evidence in favour of either model. Bayes factors below 1 indicate evidence against a difference in N400 amplitude associated with the manipulation. **C**. Posterior distribution of the difference in PNP amplitude between the violation conditions. **D**. Bayes factors under a range of priors for the PNP effect.

more plausible than smaller or larger effects, although the Bayes factors even at $1 \mu V$ were inconclusive (Fig 2D). The PNP was more positive for implausible than plausible particles, although the Bayes factor suggested only anecdotal evidence for this conclusion, $\hat{\beta} = 0.49 \, \mu V, [-0.29, 1.24], BF_{10} = 2.69$.

## Interim discussion

We compared sentence pairs where either a single verb particle or a small set of verb particles were plausible continuations given the context. We hypothesised that lower entropy would encourage commitment to a lexical prediction in the 1-particle condition, and that violating this prediction would be more costly relative to the 2+particle condition where higher uncertainty discouraged prediction. We predicted that this cost would be reflected in a larger N400 and PNP. Contrary to this prediction, evidence about the effect of entropy on both the N400 and PNP was inconclusive. The comparison of plausible versus implausible particles elicited a difference in both component in line with the typical plausibility effect observed in previous research, including research on German particle verbs [77], confirming that the experiment worked in principle, although evidence for the effect on the PNP was less conclusive than expected.

While Bayes factor evidence for the effect of entropy on the N400 was inconclusive, the magnitude of the effect was small, which could be considered consistent with the null

hypothesis of lexical access and probabilistic update accounts [42, 43, 47]. Under these accounts, the N400 at zero-probability words should not be affected by uncertainty caused by multiple activated words. One caveat to this conclusion is that, if there were truly no N400 effect, one could have expected stronger support for the null hypothesis from the Bayes factor analysis, especially since the cloze probability manipulation was able to yield strong evidential support. One reason that the Bayes factor may not have been able to distinguish between the null and alternative hypotheses is that the study was not sufficiently powered to detect or rule out an effect of entropy. While the study had a sample size of 50—larger than many ERP studies in the literature—a small number of critical trials (10 per participant) likely led to a large amount of noise in the averaged ERP data.

Bayes factor evidence for the PNP was also inconclusive, but the effect size was larger and consistent in sign with previous findings on the effect of constraint (larger PNP for violations in high-constraint settings) [32, 33, 36]. This may suggest that lower entropy led to a prediction in the 1-particle condition, eliciting a higher suppression cost once this prediction was violated. Higher entropy in the 2+particle condition may have discouraged predictions, avoiding this cost. However, again, one could have expected stronger evidence from the Bayes factor. The findings would therefore also be consistent with a null effect in line with current PNP accounts suggesting that the component should not be sensitive to entropy if the input is implausible [33, 57].

One possible explanation for the weak Bayes factors is that variability was introduced into the data by individual differences in dealing with the implausible verb particles. Particles are more syntactically and semantically ambiguous than the implausible nouns used in previous PNP research and participants may have tried to reanalyse them in different ways, while others made no attempts at revision. For example, an implausible verb particle such as **make [a story]** *__on__ could potentially be revised as a preposition **make [a story [ on the war in Iraq]]. . .**. The fact that a large posterior P600 was also elicited by the violations may indicate that such a syntactic reanalysis was attempted.

However, any conclusions from Experiment 1 are limited by a small number of critical trials, even though the sample size was relatively large for an ERP study. To test whether a larger study would provide more conclusive evidence for the PNP effect and against the N400 effect, we used the results of Experiment 1 to pre-register a second experiment with double the number of participants and more than double the number of experimental sentences.

## Experiment 2

Experiment 2 sought to replicate the findings of Experiment 1 in a larger sample, using a larger number of experimental items. Since Experiment 1 already confirmed the presence of the expected ERP components at plausible vs. implausible particles, to further increase the probability of detecting an effect, we removed the plausible conditions to create a two-condition design. This meant that the implausible conditions could be split into two rather than four lists, doubling the number of critical trials each participant would see. In line with Experiment 1, we expected that we would *not* see a difference in N400 amplitude between 1-particle and 2+particle violations, but that the PNP might be larger in the 1-particle than the 2+particle condition. Experiment 2 was pre-registered at https://osf.io/y6k2d; see also S1 Appendix.

### Methods

**Participants.**   Recruitment, inclusion, and exclusion criteria were identical to those in Experiment 1. To estimate a suitable sample size for Experiment 2, we conducted a power analysis using the data from Experiment 1. Although Bayesian analysis was used for the main

analysis, we conducted a frequentist power analysis as it was computationally lighter and serves as a quick ballpark estimate of the sample size needed for obtaining high-precision estimates [78]. The power analysis suggested that even with 300 participants we would only achieve around 70% power. Due to resource constraints, it was only feasible to collect data from around 100 participants. We therefore set a goal of 100 participants as the maximum feasible sample size and redid the power analysis, which estimated around 50% power at this sample size. While likely to be underpowered, our goal with Experiment 2 was to provide higher-precision estimates of the effects observed in Experiment 1 using an improved design, and to see to what extent doubling the size of the experiment would improve our ability to quantify evidence for our hypotheses. This information is useful to future evidence synthesis and in determining how one might approach future experiments. In total, 115 participants were recruited, 4 of whom were excluded as they did not meet the inclusion criteria. A further 11 were excluded due to technical problems with the EEG recording. This left a total of 100 participants (24 male), with a mean age of 24 years (range = 18 to 35 years, SD = 4 years).

**Materials.** Having established in Experiment 1 that the implausible particles did elicit the target ERP components relative to their plausible counterparts, for Experiment 2 we increased power by having participants see one of the two violation conditions from each item rather than one of the four total conditions. The plausible conditions were replaced by a length-matched sentence for each item that contained a separated verb-particle dependency with a plausible particle. This filler served to maintain the ratio of plausible to implausible particle verb sentences, and to serve as a sanity check. These fillers were not analysed as they were otherwise unmatched with the implausible sentences. Each participant thus saw either condition (b) *or* (d), *and* the length-matched, plausible filler from each item.

The sentences from Experiment 1 were re-used, plus 11 new sentences constructed using the same procedure as for Experiment 1. The new sentences were selected from a pool of 20 cloze-tested with 30 native German speakers (mean age 24 years, SD 5 years, range 18–38 years). To create an even number of sentences, one of the new sentences replaced the lowest-ranked of the 40 items from the original cloze test. This gave a total of 50 critical items, making a total of 50 items (25 per condition) analysed in Experiment 2. Cloze probabilities and entropy are summarised in Table 2.

In addition to the 50 target items and 50 matched fillers presented to participants, 108 general filler sentences were randomly interspersed. These were 62 of the fillers from the Experiment 1 that did not contain separated particle verbs, plus 46 new fillers, also not containing separated particle verbs. Overall, each participant saw a total of 208 sentences during the testing session. The ratio of plausible to implausible sentences was 4:1. Participants again answered a yes/no question after each sentence.

**Procedure, EEG recording, preprocessing.** The data collection procedure and preprocessing were identical to that of Experiment 1. EEG data cleaning resulted in the exclusion of 12.26% of the 5000 target trials due to artefact. A further 0.60% were excluded due to question response times over 10 seconds and 0.32% were not recorded due to experimenter error.

**Table 2. Cloze probability summary statistics for Experiment 2.**

| Condition | Cloze probability target particle Mean [95% CrI] | Difference between 1st- and 2nd-best completion Mean [95% CrI] | Entropy target particle Mean [95% CrI] |
|---|---|---|---|
| 1-particle | 0.89 [0.73, 1.00] | 0.73 [0.64, 0.84] | 0.31 [0.30, 0.32] |
| 2+particle | 0.53 [0.52, 0.55] | 0.28 [0.15, 0.38] | 0.81 [0.80, 0.81] |

1st- and 2nd-best completions refer to the highest and second-highest cloze particles at the target site.

**Analysis.**   The analysis comparing the two violation conditions for Experiment 2 was identical to that of Experiment 1. The plausible filler was not analysed as there were too many differences between these fillers and the critical items.

## Results

Fig 3A and 3B shows that the 250–500 ms window of the ERP showed a strong negative deflection in the implausible conditions relative to the plausible filler, consistent with an N400. A posterior P600 was elicited by implausible particles in the 600+ ms time window. Fig 3C and 3D also shows a positive deflection in the post-N400 period of the ERP (i.e. after 500 ms) in fronto-central electrodes consistent with a PNP, however this time in both the 1- and 2+particle conditions relative to the plausible filler, and most positive in the 2+particle condition.

**Analysis of the N400.**   As for Experiment 1, there was a small between-condition difference in N400 amplitude at implausible particles, $\hat{\beta} = -0.22\,\mu V$, 95% CrI $[-0.66, 0.22]$ Fig 4A,

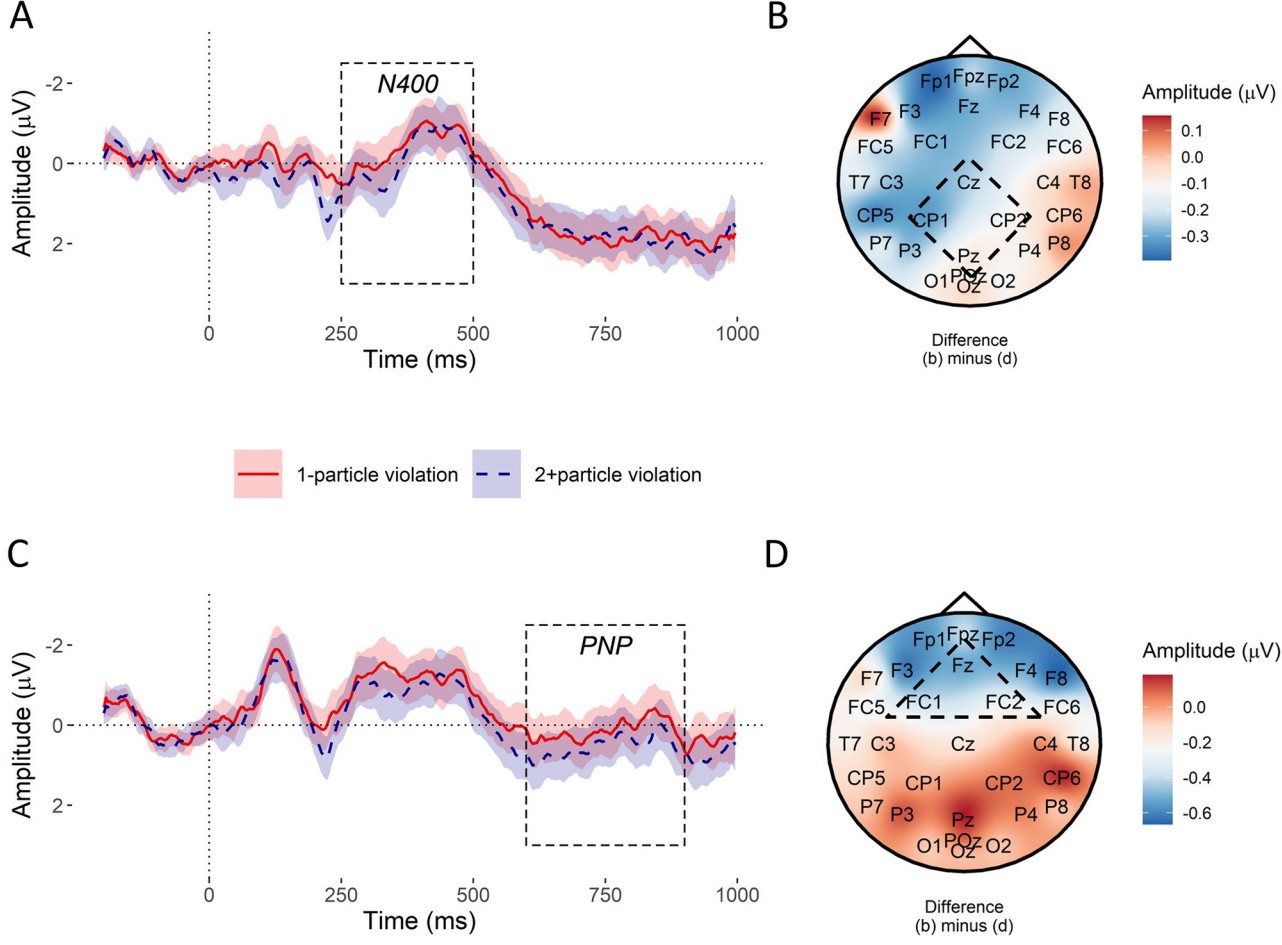

**Fig 3. Experiment 2: ERP results. A**. ERP waveforms showing an N400 in 1-particle (red) and 2+particle (blue) in the posterior region of interest (Cz, CP1, CP2, Pz). Ribbons show 95% confidence intervals calculated as for Fig 1. The pre-registered time window of the N400 is overlaid. **B**. The voltage subtraction plot of the difference in amplitude between 1- and 2+particle violations in the N400 time window indicates very little difference in amplitude between the two conditions in the posterior region of interest (dashed square). **C**. ERP waveforms in anterior (Fz, FC1, FC2) region suggest that the most positive waveform in the pre-registered PNP time window (overlaid) was for 2+particle violations. Ribbons show 95% confidence intervals. **D**. The voltage subtraction plot of the difference in amplitude between 1- and 2+particle violations in the PNP time window suggests amplitude was more positive for 2+particle violations in the anterior region of interest (dashed triangle).

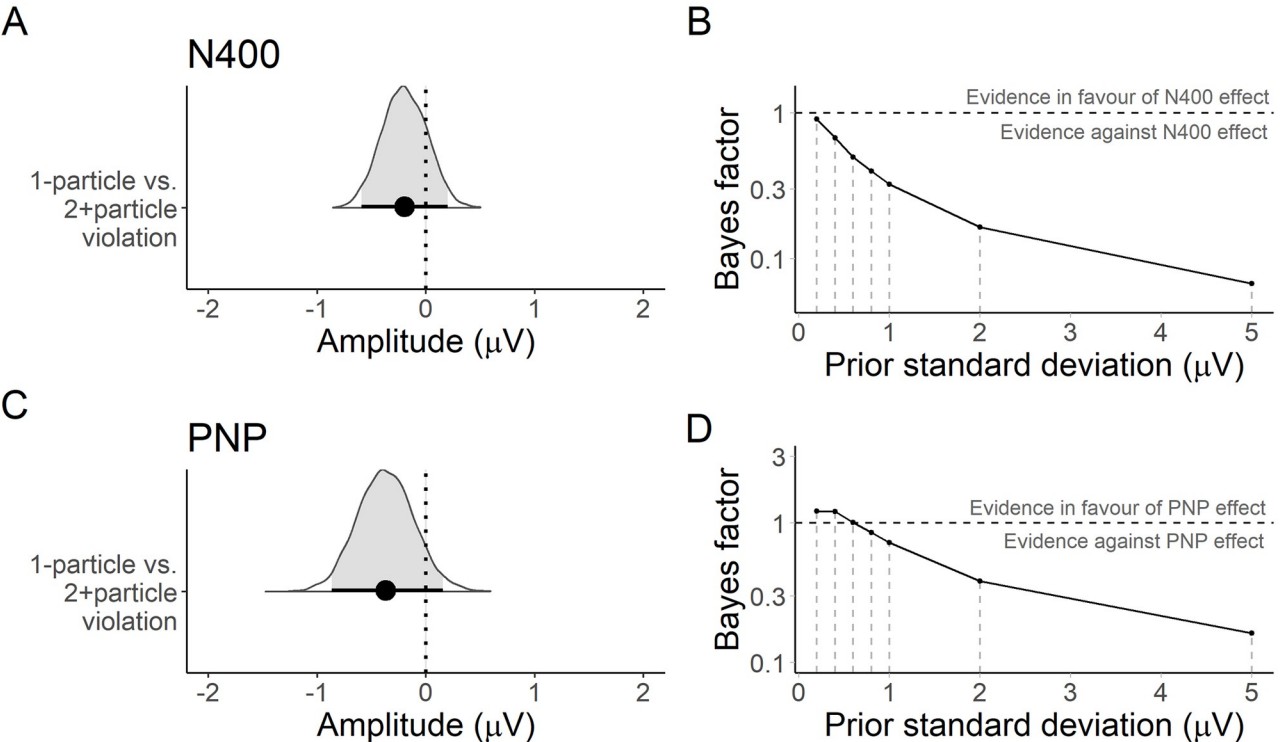

**Fig 4. Experiment 2: Posterior estimates and Bayes factor analysis for the planned comparisons. A**. Posterior distribution of the estimated difference in N400 amplitude associated with 1-particle condition vs. 2+particle violations. The mean and 95% credible interval are overlaid. **B**. Bayes factors computed under a range of priors comparing models with and without a predictor for the 1- vs. 2+particle condition manipulation. A Bayes factor of 1 indicates no evidence in favour of either model. Bayes factors below 1 indicate evidence against a difference in N400 amplitude associated with condition. **C**. Posterior distribution of the difference in PNP amplitude between the violation conditions. **D**. Bayes factors under a range of priors for the PNP effect.

but the Bayes factor at a prior of $N(0, 1)$ favoured the null hypothesis that there was no effect of entropy on the N400, $BF_{10} = 0.35$. The sensitivity analysis indicated more conclusive evidence against a priori hypothesised large effect sizes than in Experiment 1, but was still inconclusive about small effect sizes despite the increased sample size Fig 4B.

**Analysis of the anterior post-N400 positivity (PNP).** The PNP effect at implausible particles appeared to have reversed relative to Experiment 1, with a more positive waveform in the 2+ than the 1-particle condition. The posterior mean indicated a small difference in amplitude between the violation conditions, $\hat{\beta} = -0.40\,\mu V$, 95% CrI [−0.89, 0.12] (Fig 4C). However, the Bayes factor at a prior of $N(0, 1)$ favoured neither the null or alternative hypotheses about an effect of entropy on the PNP, $BF_{10} = 0.84$. The sensitivity analysis this time indicated evidence against a priori hypothesised large effect sizes, but was even more inconclusive about smaller effect sizes than for Experiment 1 (Fig 4D).

In summary, despite the relatively large amount of data and a clean manipulation of entropy, no conclusive effect on the N400 or PNP was detected. Visual inspection of the ERPs did not point to any other effects of the manipulation.

## Discussion

Experiment 2 replicated the N400 findings observed in Experiment 1, with both violation particles eliciting an N400 of a similar amplitude regardless of whether one or more than one

particle had been plausible in the context. Interestingly, the Bayes factor analysis did not become more conclusive with the increased sample size and number of trials, although the data were sufficient to rule out large effect sizes. The increased sample size and number of experimental stimuli may therefore still have been too small to yield conclusive evidence. This possibility is discussed further in the *General discussion*, as well as the relevance of the findings with respect to current models of the N400.

With respect to the PNP, the source of the reversal in effect direction relative to Experiment 1 was unclear. The difference in sign could reflect oscillation around a true mean of zero with a wide standard deviation. A true mean of zero would be in line with current accounts of the PNP, which suggests that implausible words such as our particles should not elicit a entropy-based difference in amplitude. However, the inconclusive Bayes factor may suggest that the data were too noisy to distinguish between the tested hypotheses at a priori hypothesised small effect sizes, even though sensitivity analyses indicated evidence against large effect sizes.

## General discussion

In two experiments, we investigated whether uncertainty, as measured by entropy, would influence readers' willingness to predict the lexical identity of an upcoming particle in a long-distance verb-particle dependency. Experiment 1 suggested that predictions had been made when certainty was high, based on the apparent cost of suppressing the disconfirmed sentence representation indicated by a larger anterior post-N400 positivity (PNP). However, statistical evidence was inconclusive and the effect was not replicated in the larger Experiment 2. In Experiment 2, entropy did not appear to have an effect on processing difficulty at the violations, although statistical evidence was again inconclusive. We interpret the findings as related to how readers dealt with the implausible verb particle.

### Implausible input and the N400

The lack of conclusive evidence for an entropy-based N400 difference could be interpreted as being consistent with current accounts of the N400, which would not have predicted a difference [43, 47]. This would further suggest that the highest-cloze-probability definition of constraint used in the studies on which these accounts are based is sufficiently similar to entropy, and that the distribution of co-activated words is not an important factor in explaining N400 amplitude; at least not at an implausible word. Indeed, these accounts of the N400 make no explicit predictions about how co-activated words should affect processing of a zero-probability word; however, one could have imagined that lexical access [43] or probabilistic update [47] might be dampened by the presence of co-activations if these activations consume the same resources, or create a large amount of uncertainty in the system. Co-activated words have in fact been observed to modulate amplitude of the N400 at *plausible* but unexpected words [38, 42, 47, 79]. One contribution of the current study may therefore be to confirm the implicit assumption of current N400 accounts that uncertainty caused by competing lexical representations does not affect N400 amplitude if the input is implausible.

One caveat to any conclusion about the N400 in the current study is the inconclusive Bayes factor analysis at small effect sizes. If the N400 is truly not affected by constraint, then with sufficient data, one should have expected a Bayes factor at least weakly in favour of the null hypothesis. While the data were sufficient to find evidence against a priori hypothesised large effect sizes, the inconclusive evidence against small effect sizes, even in the larger Experiment 2, raises several possibilities: first, that some property of the verb-particle violation differs to the types of violations in which the N400 is usually studied and that this difference elicits a less clear-cut effect on the N400; second, given a less clear-cut effect, that an even larger sample

size would have been needed to provide conclusive evidence against an entropy effect; or third, that implausible words really did elicit a small difference in N400 amplitude associated with entropy, but that the difference was too small to yield conclusive evidence at the current sample size. We discuss the properties of the verb-particle dependency next in relation to the PNP, in which the interaction of plausibility and constraint (a proxy for entropy) has been directly tested in previous research.

With respect to sample size, we again used a frequentist power analysis to gain a rough estimate of the sample size that would be needed for a hypothetical, future experiment using the same experimental design and assuming that the effect was the same size observed in Experiment 2 [80]. This prospective power analysis suggested that thousands of participants would be needed to achieve even 30% power, strongly suggesting that the above questions are likely not answerable with this particular design. Compare this to the power analysis conducted using the lower-precision estimates from Experiment 1, which vastly overestimated that we could achieve around 50% power with just 100 participants.

## Entropy, implausibility, and the PNP

Previous findings suggest that constraint—similar to entropy—does not modify the PNP when the input is implausible [33, 57]. The lack of conclusive evidence for an entropy effect at implausible particles in the current study may therefore be consistent with the hypothesis that the PNP reflects only *successful* update of a mental sentence representation triggered by plausible input [33]. However, if the PNP is not elicited by implausible input, we should have expected conclusive evidence in favour of the null hypothesis, especially in the larger Experiment 2. It is therefore possible that something about the violation particle was different to the types of violations in which the plausibility effect on the PNP has previously been studied. Specifically, the implausible particle violates expectation at a purely semantic level of representation, since any syntactic expectation for 'a particle' was satisfied. In contrast, previous PNP studies have typically used verb-noun dependencies, where an implausible noun violates expectations at a number of levels of representation (e.g. animacy, thematic role, semantic). This raises the possibility that implausibility affects predictive processing cost in a graded way, with 'lighter' implausibility being simply less costly.

Alternatively, lighter violations may be more amenable to repair. In the current study, repair may have been aided by the fact that the verb-particle construction denotes a more abstract concept than a noun, which may allow attempts to update the semantic representation. For example, *carried with* is not a licensed verb-particle combination, but one could perhaps more easily invent a situation to accommodate it than to accommodate an implausible noun. Moreover, German particles are identical to prepositions and thus readers may have attempted to revise the particle as a preposition, e.g. *carried his lecture [with determination] on*. The large P600 in both experiments may suggest that such reanalysis was attempted [33, 46, 52, 54, 81–84].

In further support of a role of dependency type in modulating whether implausibility elicits the PNP, visual inspection of the ERPs in both experiments suggested that implausible particles did elicit a more positive waveform in the PNP window than plausible particles. Previous research in implausible nouns suggests that this should not have been the case [33, 57]. However, the plausible and implausible particles in both experiments were not matched for frequency or lexical identity, and so this finding requires future investigation in a more controlled experiment. In summary, one possible explanation for the inconclusive findings in both the PNP and N400, despite the large sample size, is that the level of representation at

which expectations are violated may be an important factor in using ERPs to measure how entropy affects predictive processing.

## Entropy and the P600

While the P600 was not a focus of the current experiments and so was not analysed, its visual appearance in Fig 3 (Experiment 2) suggests that it was not affected by entropy. This contrasts with previous studies of anomalous words, where violations of strong constraint elicited larger P600s than violations of weak constraint [33]; for constraint effects at unexpected but still plausible words, see [85, 86]. Kuperberg and colleagues propose that stronger constraint makes the discrepancy between expected and received input larger, leading to a larger P600. This could suggest that the meaning discrepancy in the current study was not large enough to elicit a constraint effect.

However, there is another interesting possibility: that the violation occurred at different levels of representation between studies. Kuperberg and colleagues violated the animacy and thematic role constraints of expected nouns. In contrast, our particles violated only semantic constraints: there are no thematic roles or animacy features associated with particles, and although they could be reanalysed as prepositions, the morphosyntactic form of either a particle or preposition was not violated. Our P600 therefore contrasts with accounts of the P600 as a discrepancy between lexical expectations and expected morphologic form [81, 85, 87], as well as with accounts associating it with syntactic processing [51, 54, 88]. It is perhaps more consistent with accounts of the P600 as reflecting syntactic reprocessing attempts [89], general error detection and reanalysis process [83, 90], or integration processes [52, 55, 56]. We leave to future research the question of whether the difference between our and Kuperberg and colleagues' violations suggest that violating different levels of representation affect the P600 differently.

## Alternative accounts of ERP findings at the particle

Based on previous findings, we have attributed our inconclusive results to the difficulty of interpreting ERPs elicited by implausible words rather than to evidence that predictions were not made based on previous findings in particle verbs and similar constructions [3–5, 91]. However, a non-prediction account is also compatible with the current findings. An integration account would propose that the lexical identity of the particles was not preactivated, but rather that processing was initiated once the particle was encountered in the input [92–94]. Being equally unintegrateable into the sentence, an equal amount of processing difficulty would then be reflected in the amplitude of the ERP components. An integration account could be contradicted by examining ERP differences earlier in the sentence; for example, by examining whether working memory load differed in line with whether a prediction had been made or not, as indexed by the left anterior negativity (LAN) or sustained anterior negativity (SAN). However, it is unclear whether the LAN is sensitive to working memory load associated with lexical ambiguity as it is more often studied in relation to syntactic structure building [12–14], although lexical ambiguity was found not to play a role in one study [91]. An investigation of these components is beyond the scope of the current article, but would be worthy of future examination to further disentangle prediction from non-prediction accounts.

A second alternative account of our findings is that of Piai et al. (2013), who concluded that the lexical representations of verbs that take particles—rather than of the particles—are maintained in working memory to facilitate their retrieval once the particle is encountered. Since the verbs should have been equally retrievable, this would explain why no differences in ERP amplitude were observed at the particle site. Our findings would also support the idea that a

syntactic placeholder for a particle was posited, in the same way that gaps are posited in filler-gap dependencies [12–14], but that the lexical features of the required particle were not activated until the particle was encountered. As mentioned, there is some evidence from reading time studies to suggest that entropy may influence the preactivation of long-distance lexical information [3–5], but for now, ERP evidence remains elusive.

## Conclusions

Using long-distance particle verb dependencies in German, we tested the hypothesis that uncertainty may affect readers' willingness to commit to a specific sentence meaning. Long-range lexical predictions may be advantageous to interpreting an event in German particle verb dependencies where the full verb form can be split across almost an entire sentence. However, we were unable to find conclusive evidence that uncertainty affected lexical predictions, even in such a high-stakes context where predictions would have been particularly useful. This result could have been due to the comparison of ERPs at implausible target words; however, if this was the case, it is surprising that evidence *against* an effect of uncertainty—as measured by entropy—was so inconclusive given clear predictions made by the literature about implausible input, and given that the study design was sufficient to demonstrate evidence for a standard cloze probability effect on the N400.

We did find evidence, however, that manipulating entropy was unlikely to produce changes in amplitude at large a priori hypothesised effect sizes. Thus, our findings do not rule out the possibility that an entropy effect might be present but small in magnitude. A small-magnitude entropy effect could suggest that the semantic violation caused by an implausible verb-particle combination has less clear consequences for processing cost than the violations at multiple levels of representation caused by anomalous nouns in previous studies. This raises questions such as whether simpler violations are less costly or whether they are more amenable to repair. The properties of particle verb dependencies therefore present an interesting way to extend our understanding of probabilistic processing cost.

## Supporting information

**S1 Appendix. Detail on pre-registrations.**
(PDF)

**S2 Appendix. Pre-registered single electrode analyses for Experiment 1.**
(PDF)

## Acknowledgments

We thank Johanna Thieke, Romy Leue, Chiara Tschirner, and Alexandra Lorson for their help with designing the stimuli and collecting the data.

## Author Contributions

**Conceptualization:** Kate Stone, Titus von der Malsburg.

**Data curation:** Kate Stone.

**Formal analysis:** Kate Stone, Shravan Vasishth, Titus von der Malsburg.

**Investigation:** Kate Stone.

**Methodology:** Kate Stone, Titus von der Malsburg.

**Project administration:** Kate Stone, Titus von der Malsburg.

**Supervision:** Shravan Vasishth, Titus von der Malsburg.

**Visualization:** Kate Stone.

**Writing – original draft:** Kate Stone.

**Writing – review & editing:** Kate Stone, Shravan Vasishth, Titus von der Malsburg.

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
