## [Decision Letter · Decision Letter 0]

13 Dec 2021

PONE-D-21-31739Does entropy modulate the prediction of German long-distance verb particles?PLOS ONE

Dear Dr. Stone,

Thank you for submitting your manuscript to PLOS ONE. After careful consideration, we feel that it has merit but does not fully meet PLOS ONE’s publication criteria as it currently stands. Therefore, we invite you to submit a revised version of the manuscript that addresses the points raised during the review process.

I would invite the authors to carefully consider all the suggestions advanced by both Reviewers. I concur that the paper in its present form is well-written and worth to be published. However, I am in line with Reviewer #1 that "this paper uses statistics from the future, but ERP analysis/visualisation from the past". Following the Reviewers' points the paper can be largely improved.

We look forward to receiving your revised manuscript.

Kind regards,

Nicola Molinaro, Ph.D.

Academic Editor

PLOS ONE

“Kate Stone was supported by a doctoral scholarship from the Potsdam Graduate School.”

“Kate Stone was supported by a doctoral scholarship from the Potsdam Graduate School.”

We note that you have provided information within the Acknowledgements Section. Please note that funding information should not appear in the Acknowledgments section or other areas of your manuscript. We will only publish funding information present in the Funding Statement section of the online submission form.

“Kate Stone was supported by a doctoral scholarship from the Potsdam Graduate School.”

Reviewers' comments:

Reviewer's Responses to Questions

**Comments to the Author**

1. Is the manuscript technically sound, and do the data support the conclusions?

Reviewer #1: Yes

Reviewer #2: Yes

2. Has the statistical analysis been performed appropriately and rigorously? 

Reviewer #1: Yes

Reviewer #2: Yes

3. Have the authors made all data underlying the findings in their manuscript fully available?

Reviewer #1: Yes

Reviewer #2: Yes

4. Is the manuscript presented in an intelligible fashion and written in standard English?

Reviewer #1: Yes

Reviewer #2: Yes

5. Review Comments to the Author

Reviewer #1: Does entropy modulate the prediction of German long-distance verb particles?

Stone et al. present an ERP study on an interesting topic, the role of uncertainty in lexical prediction (i.e. commitment to a specific word). They approach it by cleverly exploiting long-distance verb particles in German. Results of the initial experiment turned out inconclusive, so the authors (laudably) ran a replication with a large sample size. In the end, the results were still inconclusive (statistically speaking at least, in practice _I think_ we can at least conclude there was not a lot of evidence for the hypothesis). However, the authors fully embraced this uncertainty using sophisticated and comprehensive Bayesian analyses. They also make some interesting suggestions as to why the results turned out the way they did.

Overall, I think this is a good study: smart design, rigorous methods, appropriate (in)conclusions. I have no major comments. To my mind, it could in principle be published as-is. However, I do have _some_ questions and comments that the authors may want to engage with. Hopefully (some of) it will be useful in finalizing the manuscript, in case of some very minor revisions.

Not-too-major comments:

1. Zero-probability words

A key assumption in the design is that the low and high uncertainty condition, implausible particles are “zero probability” – defined in a cloze task – and hence equally unlikely. As such, any potential difference can be attributed to the uncertainty, not predictability.

However, we know that predictability effects are logarithmic: comprehenders are sensitive to subtle differences between seemingly small probabilities (e.g. 0.005 vs 0.0005). Words with such low probabilities will be practically never produced in a cloze norming task, and hence are seemingly equally unlikely, although they are not in fact of equal predictability.

One instance where this may apply is in the amenability to repair. The authors write that:

“German particles are identical to prepositions and thus readers may have attempted to revise the particle as a preposition, e.g .”carried his lecture [with determination] on”. The large P600 in both experiments may suggest that such reanalysis was attempted” (735-737)

Could it be that some “implausible” articles are more open to such alternative interpretations than others? If so, this would arguably reflected in their predictability: implausible words that can be “saved” with alternative interpretations will be less unlikely. If the materials were English, this would be easy enough to assess: use a modern language model to quantify the unpredictability of the implausible articles. Perhaps this could capture additional variance, and strengthen the inference. For German, I am not sure if sufficiently competent LMs exist though (if so, they will probably be on the modelhub at hugginface though).

To be clear, these are just suggestions and I don’t think any such additional analyses are required for acceptance. However, I was struck by the fact that the authors so uncritically use the notion of “zero probability”, while also appealing to the concept of probabilistic/graded pre-activation, and error-based accounts of N400 (e.g Fitz & Chang, 2019), where this error is based on a softmax that will almost never return a predictability of absolutely zero?

2. Measurement and use of entropy

The authors extensively appeal to the concept of entropy, which (rightly so, IMO) they suggest is a better formalization of (inverse) constraint than highest cloze probability. But if I’m not mistaken, they only use a categorical distinction between “number of plausible particles” in their statistical comparison. Why is this? Why not use the empirical entropy? Surely the entropy from the cloze will be imperfect, but it would contain more information about the distribution than the simple distinction between 1-particle vs 2+-particle?

I’m sure the authors have good statistical reasons for this, but maybe they can give them?

3. Uncertainty and the notion of lexical prediction

This is point is more of a conceptual question than a comment.

The authors investigate the conditions under which comprehenders make a lexical prediction, here defined as an all-or-none commitment to a specific word. Since such a commitment is risky and can lead to “costly misprediction”, the hypothesis was that comprehenders only make such a commitment when the entropy is low.

Now, this all seems perfectly reasonable, and well in line with how lexical prediction has long been conceptualized in psycholinguistics (e.g. van Petten and Luka, 2012). But I was wondering: is there any evidence that comprehenders actually make such commitments? I’m asking because in the introduction, the authors seem to take for granted that ‘lexical prediction’ (i.e. commitment) and probabilistic/graded pre-activation both exist in parallel. But is there any evidence for this (ignoring visual world tasks, that enforce commitment)? Isn’t the evidence equally consistent with the idea that all predictability effects can be attributed to a kind of graded preactivation (with more or less certainty)? And relatedly, if graded/probabilistic prediction is all there is, would the authors still expect an effect of entropy, over and above surprisal? Why/why not?

Even-more-minor comments:

a) EEG channel selection.

The authors use an ROI to define the N400. This apparently diverges from their preregistration but it seems a good choice to me. The selected channels also seem reasonable. However, what is the motivation for using these particular channels? Is it a prior study, if so maybe they can give it?

b) EEG visualization and analysis.

While I was very impressed with the statistics, I was a bit disappointed by the presentation of the EEG data and results. EEG results only convey point estimates of the mean, no errorbars or other visualisations of statistical consistency. For instance, looking at the red line and blue dashed line in Figure 1a, the plot is equally consistent with a very strong effect in the expected direction, as with no effect. I understand that the authors’ statistical analysis is too sophisticated to be accurately conveyed by any single error bar. However, as currently presented the EEG visualisations are really not that informative.

Again, this is not something that _must_ be fixed before acceptance. However, I did want to say that as a reader, I get the impression that this paper uses statistics from the future, but ERP analysis/visualisation from the past… Maybe something to consider in future projects?

Reviewer #2: This paper presents a study with two experiments (the second being a replication of the first) examining the effect of semantic predictability in sentence processing: Do speakers of German predict specific particles in particle-verb constructions, where quite a lot of intervening material can come between the two elements? They compared sentences where either a single particle was semantically plausible versus sentences where more than one particle was plausible. The critical contrast was in cases where a violation occurred, which would ultimately both elicit the ERP components of interest (N400 and PNP). The main question was whether the magnitude of these effects would differ in each case (and in the case of the PNP, whether it would occur at all). On one hand, both may elicit an equal effect—a potentially meaningful null effect, indicating that components such as the N400 are not sensitive to entropy. On the other hand, there may be a difference, indicating both that the N400 is sensitive to entropy, and that such distributional facts are being used by language comprehenders.

First, some positive notes. To my knowledge, the authors are making excellent use of statistical tools to analyze the results of their studies, and the questions that motivate this study are of general interest. At the end of the day, however, the results of this study do not bring us any closer to finding an answer to the very good questions that the authors pose. As demonstrated by the Bayes Factor analysis, the findings are inconclusive. No difference was observed between the critical conditions in either the N400 or PNP, and we cannot have confidence that this is a true null effect. The authors do an excellent job of carefully presenting and discussing this inconclusiveness. I think this is a model for how a paper should handle such results.

What gives me particular pause is that the lack of a conclusive result was highly predictable. The authors report that a power analysis was run prior to Experiment 2 based on the results of Experiment 1, and yet, they still ran a highly underpowered study. Only 1/3 of the number of participants to achieve 70% power were run (Side note: what is the power for the study that was actually run? That is, the power for a 100 person study? This is not reported, but should be). The authors make statements (e.g. in the abstract) that the results were “surprisingly unclear”. I actually do not think, per the authors own power analysis, that this unclarity is at all surprising. Quite the opposite, it was expected. I think this should be more directly addressed.

Generally I think this paper would benefit by being re-written in a way that does not lead to any expectation that the main research question will be answered. Perhaps more of a methods paper with a deeper literature review on the components of interest. There are some good hints of what this might look like in the discussion and conclusion. For example, focusing the paper around the interpretation of ERP components rather than around whether language comprehenders use entropy to make predictions. This would be a more appropriate framing for a paper that lacks interpretable results and cannot actually answer those questions.

Additional Comment:

Study 2 says 115 were excluded, 16 were excluded (4 for not meeting inclusion criteria, 12 for technical issues), but then 100 were included in the analysis. There must be a mistake somewhere, because 115 minus 16 is 99, not 100.

---

## [Author Response · Author response to Decision Letter 0]

16 Mar 2022

Please see attached file "Response_to_reviewers.pdf"

---

## [Decision Letter · Decision Letter 1]

18 Apr 2022

Does entropy modulate the prediction of German long-distance verb particles?

PONE-D-21-31739R1

Dear Dr. Stone,

We’re pleased to inform you that your manuscript has been judged scientifically suitable for publication and will be formally accepted for publication once it meets all outstanding technical requirements.

Kind regards,

Nicola Molinaro, Ph.D.

Academic Editor

PLOS ONE

Reviewers' comments:

Reviewer's Responses to Questions

**Comments to the Author**

1. If the authors have adequately addressed your comments raised in a previous round of review and you feel that this manuscript is now acceptable for publication, you may indicate that here to bypass the “Comments to the Author” section, enter your conflict of interest statement in the “Confidential to Editor” section, and submit your "Accept" recommendation.

Reviewer #1: All comments have been addressed

Reviewer #2: All comments have been addressed

2. Is the manuscript technically sound, and do the data support the conclusions?

Reviewer #1: Yes

Reviewer #2: Yes

3. Has the statistical analysis been performed appropriately and rigorously? 

Reviewer #1: Yes

Reviewer #2: Yes

4. Have the authors made all data underlying the findings in their manuscript fully available?

Reviewer #1: Yes

Reviewer #2: Yes

5. Is the manuscript presented in an intelligible fashion and written in standard English?

Reviewer #1: Yes

Reviewer #2: Yes

6. Review Comments to the Author

Reviewer #1: Thanks for the insightful responses and revisions, they have addressed all my questions.

Reviewer #2: I greatly appreciate the time that the authors took to address my initial comments. To recap, I had two main issues that were mainly related to the rhetorical structure: (i) the way the still under-powered replication was presented, and (ii) modifying the expectations of the reader with regards to the still inconclusive nature of the results earlier in the paper. I feel that both of these comments have been adequately addressed in the revisions to the manuscripts and well-motivated in the responses as well. Again, I thank the authors for taking these to heart. I completely agree that this should not go into the file drawer, and as I said in my initial review, I find it to be a model of how to present, analyze, and discuss inconclusive results.

---

## [Editor Report · Acceptance letter]

18 Jul 2022

PONE-D-21-31739R1 

Does entropy modulate the prediction of German long-distance verb particles? 

Dear Dr. Stone:

I'm pleased to inform you that your manuscript has been deemed suitable for publication in PLOS ONE. Congratulations! Your manuscript is now with our production department. 

Kind regards, 

on behalf of

Dr. Nicola Molinaro 

Academic Editor

PLOS ONE